# From Past to Present: The Link Between Reactive Oxygen Species in Sperm and Male Infertility

**DOI:** 10.3390/antiox8120616

**Published:** 2019-12-03

**Authors:** Ana Izabel Silva Balbin Villaverde, Jacob Netherton, Mark A. Baker

**Affiliations:** 1Independent researcher, São Paulo 13000-000, Brazil; aivillaverde@hotmail.com; 2Department of Biological Science, University of Newcastle, Callaghan, NSW 2308, Australia; jacob.netherton@newcastle.edu.au

**Keywords:** dihydroethidium, lucigenin, luminol, tetrazolium salts, NADPH oxidase, cytochrome reductases

## Abstract

Reactive oxygen species (ROS) can be generated in mammalian cells via both enzymatic and non-enzymatic mechanisms. In sperm cells, while ROS may function as signalling molecules for some physiological pathways, the oxidative stress arising from the ubiquitous production of these compounds has been implicated in the pathogenesis of male infertility. In vitro studies have undoubtedly shown that spermatozoa are indeed susceptible to free radicals. However, many reports correlating ROS with sperm function impairment are based on an oxidative stress scenario created in vitro, lacking a more concrete observation of the real capacity of sperm in the production of ROS. Furthermore, sample contamination by leukocytes and the drawbacks of many dyes and techniques used to measure ROS also greatly impact the reliability of most studies in this field. Therefore, in addition to a careful scrutiny of the data already available, many aspects of the relationship between ROS and sperm physiopathology are still in need of further controlled and solid experiments before any definitive conclusions are drawn.

## 1. Introduction

The history of the relationship between reactive oxygen species (ROS) and spermatozoa starts with some fundamental experiments conducted by John MacLeod in 1943 [1]. Considering the knowledge accumulated till the present day, it is assumed that ROS, in high enough concentrations, can trigger peroxidative damage by the generation of reactive aldehydes, which are detrimental to cell function. This perception was demonstrated in different ways. However, in most studies, the negative effect of ROS on sperm quality was observed following external addition of ROS or exposure to ROS-generating in vitro systems, thus diverging from a physiological scenario. In this review, we discuss the many challenges in this field, including the various pitfalls associated with the techniques used for measuring ROS, which make it difficult to ascertain whether these compounds are a major factor contributing to male infertility or just metabolites playing a passive role. Nonetheless, novel and better methods for measuring ROS together with the current understanding of the pathways associated with peroxidative damage will certainly allow new insights into the involvement of oxidative stress in sperm function and male infertility.

## 2. The Foundation of the Link between ROS and Human Sperm

The earliest citation of the presence of ROS in spermatozoa comes from the laboratory of John MacLeod [1]. In 1943, MacLeod [1] decided to test the prevailing knowledge that the metabolism of human spermatozoa was exclusively dependent on glycolysis and that oxygen consumption was “*being of such small magnitude that it could not properly be interpreted as true respiration*” (cited from MacLeod [1]). Therefore, to investigate the existence of mitochondrial activity, MacLeod [1] used methylene blue as a redox sensor and observed that human sperm can reduce either glucose or succinate. In the case of succinate, the reduction of methylene blue is likely a consequence of the production of FADH_2_ in the presence of succinic dehydrogenase (or electron transport chain Complex II), an enzyme of the mitochondrial respiratory complex that oxidizes succinate into fumarate. In addition, the oxidation of *p*-phenylenediamine by sperm cells was also observed in MacLeod’s experiments, indicating the presence of cytochrome b, cytochrome c and cytochrome c oxidase. As such, this was the first evidence that sperm cells have indeed mitochondrial activity, or as better phrased by MacLeod, that they present an “*active cytochrome*” system. 

Following these first observations, MacLeod [1] further examined the impact of high oxygen levels on sperm cells. For this purpose, he incubated human sperm in a 95% oxygen environment at 38 °C. Under these conditions, a drastic reduction in sperm motility occurred over time, which was completely prevented when the experiment was repeated in the presence of catalase, an enzyme that converts hydrogen peroxide (H_2_O_2_) into water and oxygen [1]. The notion here is that when forced to use oxidative phosphorylation, a toxic by-product is created in the form of H_2_O_2_. In fact, as revealed later by others, up to 0.2% of the oxygen used during mitochondrial respiration undergoes incomplete reduction, forming superoxide anion (O_2_•^−^), which quickly reacts (dismutation) producing H_2_O_2_ [2] (for details see Figure 1). The latter can be fully reduced to water or may form oxygen radicals, such as the hydroxyl radical (•OH), that are subsequently detrimental to sperm. Thus, the fundamental concept that ROS can negatively affect spermatozoa function was laid [3,4,5,6].

MacLeod [1] reasoned that spermatozoa were the major source of ROS, but later reports showed that leukocytes within sperm samples, a common feature among human ejaculates, were also involved in ROS production [7,8,9]. Leukocytes contain an NADPH-oxidase (NOX) that catalyses the production of O_2_•^−^ by the oxidation of NAD(P)H [10]. The O_2_•^−^ is then used to generate a wide range of reactive oxidants, with the main purpose of killing invading microorganisms [10]. However, this enzyme is so active that spermatozoa can be immobilised by as little as 6 × 10^5^ stimulated leukocytes [8].

Motivated by the observations on the NOX activity of leukocytes, Whittington and Ford decided to reinvestigate the impact of high oxygen levels (i.e., 95% O_2_ and 5% CO_2_ versus 95% N_2_ and 5% CO_2_) using MacLeod’s methodology. However, this time, sperm samples were freed of leukocytes following purification by Dynabeads [11]. Of interest, the leukocyte-free sperm populations were less affected by the high oxygen tensions and remained motile for over 6 h, showing only a reduction in curvilinear velocity. This finding clearly raises the question of whether sperm produce enough ROS to cause any significant cell damage.

### 2.1. Spermatozoa and Their Susceptibility toward ROS

Regardless of the ROS source, the work developed by John MacLeod inspired a generation of andrologists to look at the susceptibility of spermatozoa towards these metabolites. Arguably, Thaddeus Mann was one of the first to realize the clinical significance of this association. In a landmark paper with Roy Jones and Dick Sherins, published in 1978, sperm cells showed motility loss when exposed to either exogenously introduced fatty acid, previously treated with UV light, or peroxidation of endogenous sperm phospholipids, induced by ascorbate and ferrous sulphate [12]. Both treatments are known to catalyse the oxidation of unsaturated fatty acids, forming unsaturated aldehydes such as acrolein, malonaldehyde (MDA) or 4-hydroxy-2-nonenal (4-HNE) [13]. Indeed, MDA production was confirmed by Jones et al. [12] using the thiobarbituric acid reacting substances test (TBARS). The active aldehydes formed by ROS can react with proteins through a Michael-type addition, particularly with the sulfur atom of cysteine, the imidizole nitrogen of histidine and the amine nitrogen of lysine [14]. Importantly, the toxic effect exerted by the covalently bounded aldehydes depends on the role of the adducted residue (e.g., protein structure and/or function) [15].

Currently, evidence shows that 4-HNE, a by-product of lipid peroxidation, can impair the activity of enzymes from the glycolytic and oxidative phosphorylation pathways, such as glyceraldehyde-3-phosphate-dehydrogenase (GAPDH) [16] and cytochrome-c oxidase [17]. In addition, 4-HNE has the potential to form adducts with A-kinase anchor protein 4 (AKAP4) and dynein heavy chain [18], two proteins involved in sperm motility [19,20]. These findings could explain the two main points observed by Jonas et al. [12], i.e., that sperm motility is impaired by reactive aldehydes and that necrozoospermic samples have higher MDA levels. Regardless of the adducts formed, the work developed by Jones et al. [12] appears to have sparked a major interest in the field of “ROS and defective spermatozoa” and, for the first time, offered a mechanism into why men may become infertile.

Following these initial observations, other reports have clearly confirmed that both ROS and aldehydes are detrimental to sperm function. However, something easily overlooked is the fact that most of the approaches within this theme use an exogenous source of ROS/aldehyde or force spermatozoa to generate ROS. For instance, much of the early work, reporting sperm motility loss, involved the use of exogenously added ascorbate plus ferrous ion [12,21,22,23,24], which induces the production of O_2_•^−^, H_2_O_2_ and •OH. Other examples include the addition of exogenous xanthine–xanthine oxidase [25,26,27,28,29], H_2_O_2_ [24,26,30,31], glucose and glucose oxidase [30], nitric oxide radical [32], menadione [24,33] and unsaturated aldehydes (acrolein, 4-HNE and MDA) [18,34,35,36]. All of these methods were shown to be detrimental to spermatozoa by the same pathway investigated by Jones et al. [12], in which the aldehydes formed by oxidation of unsaturated fatty acids lead to inhibition of sperm motility.

Although the external addition of the aforementioned compounds clearly affects sperm function, what remains a challenge to the field is the significance of this finding when we consider only the level of lipid peroxidation that occurs spontaneously in vitro and, most importantly, in vivo. Early studies with rabbit, mouse and human sperm have shown that spontaneous lipid peroxidation, based on MDA measurements, occurs at a slow rate, and factors such as temperature, oxygen tension and medium composition may greatly interfere [37,38,39,40]. In a work performed with stallion, the percentage of sperm naturally expressing 4-HNE increased from 53% to 86% over a 24 h incubation period under aerobic conditions [41]. In accordance, after 24 h, a slight increase in lipid peroxidation was detected in human sperm using the probe C11-BODIPY(581/591) [34]. These increments in lipid peroxidation were accompanied by a loss in sperm motility, which may limit sperm lifespan within the female tract [37,38,39,41]. Of interest, the lifetime of human sperm (i.e., time for complete loss of motility) was shown to be highly correlated with their level of superoxide dismutase (SOD) activity (*r* = 0.97) [40], strongly suggesting that peroxidation involving O_2_•^−^ may play a major role in motility loss over time. Nevertheless, Aitken et al. [42] observed that SOD levels on both low- and high-density sperm populations, following Percoll separation, were negatively correlated with total motility after 24 h of incubation (*r* = −0.303 and *r* = −0.338, respectively). Although SOD activity was measured by different methods, one with acetylated ferricytochrome [40] and the other with lucigenin [42], this might not account for the contrasting data.

### 2.2. Polyunsaturated Fatty Acids Quantity and Sperm Susceptibility

It is quite clear that sperm motility is affected by ROS despite their source, and most likely this is related to lipid peroxidation. The question that arises from this observation is: Why are spermatozoa vulnerable in this regard? One argument put forward is that “*mammalian spermatozoa membranes are very sensitive to free radical-induced damage*” (cited from [43]) due to their high level of polyunsaturated fatty acids (PUFA). In whole ejaculates, the overall PUFA content in spermatozoa is between 36% and 39%, while in Percoll-purified sperm, this level reaches 48–52% of the total fatty acids [44]. This clearly demonstrates that spermatozoa are within the range of other PUFA-enriched tissues such as brain, retina and placenta (around 35%, 37% and 44% PUFA, respectively) [45,46,47]. In general, the hydrogen of the bisallylic methylene group (i.e., between two double bonds) has a weak bond energy (around 75 Kcal/mol) when compared to the ones present in allylic methylene groups and methylene groups that show bond dissociation energy of approximately 88 and 101 Kcal/mol, respectively [48,49]. Considering that bisallylic carbons are only present in PUFA, this intrinsic characteristic makes them more prone to peroxidation than monounsaturated and saturated fatty acids, therefore increasing the susceptible of PUFA-rich membranes such as those of sperm cells.

The most abundant PUFA in Percoll-purified human sperm was shown to be docosahexaenoic acid (DHA, 22:6*n*-3), followed by arachidonic acid (AA, 20:4*n*-6) and linoleic acid (LA, 18:2*n*-6), with around 34.5%, 10.5% and 6.5% of the total fatty acids, respectively [44]. Likewise, DHA is also the predominant PUFA in ruminant sperm cells [50,51]. High concentrations of DHA are also found in rod photoreceptors [52] and synaptosomes [53], where they likely modulate membrane properties, including “fluidity”, flip-flop, membrane fusion and vesicle formation (reviewed by [54]). These properties are also known to be important for sperm function, thus making DHA a crucial membrane component for this cell type. To demonstrate its importance, Roqueta-Rivera and colleagues [55] showed that male mice depleted of the delta-6 desaturase enzyme, which participates in the synthesis of AA and DHA, have impaired fertility that can only be restored upon oral supplementation of DHA. Of interest, DHA presents higher oxidisability when compared to LA and AA due to their greater amount of bisallylic methylene groups [56]. Therefore, the benefits of having a singular high amount of DHA come at the expense of making sperm even more susceptible to lipid peroxidation.

The main α, β-unsaturated aldehyde formed by non-enzymatic oxidation of DHA is 4-hydroxy-2-hexenal (4-HHE), whereas the n-6 PUFA (e.g., LA acid and AA) generate 4-HNE [57]. These 4-hydroxyalkenals are very reactive and may serve as second toxic messengers, thus mediating the detrimental effects of oxidative stress upon sperm cells. For instance, even at femtomolar concentrations, 4-HHE is capable of inducing transition pore opening in mitochondria [58], which could be responsible for sperm motility loss and apoptotic changes [59,60]. However, despite its likely importance, the level of induced or spontaneous in vitro production of 4-HHE has never been examined in human sperm cells, yet it would theoretically be a more sensitive marker of oxidative stress. In contrast, 4-HNE has already been assessed and associated with a concomitant motility loss in stallion and human sperm [18,34,61,62].

Another by-product of the non-enzymatic oxidation of both n-3 and n-6 PUFA is the 3-carbon aldehyde MDA [63]. Although less toxic than 4-HNE, MDA is often used as a biomarker of lipid peroxidation due to its facile reaction with thiobarbituric acid. Nevertheless, the reliability of the TBARS test has been questioned by many, with one article stating that the “*MDA assay is not able to provide valid analytical data for biological samples due to its high reactivity and possibility of various cross-reactions with co-existing biochemicals*” [64]. Certainly, MDA levels have been found to be higher within infertile sperm [23,65,66,67], but the TBARS test has been used in all cases and, hence, further work is necessary to confirm these findings.

### 2.3. Leukocytes and Their Contribution to ROS Generation

Throughout the history of the relationship between ROS and sperm function, many have argued in favour of the hypothesis that the presence of seminal leukocytes is a confounding factor [68,69,70,71]. In this regard, reports correlating the number of white blood cells (WBC) within ejaculates and sperm dysfunction have shown both positive [68,69] and negative [72,73] correlations. In an intriguing study run by Harrison et al. [74], fertile men (i.e., fathered within 12 months) showed great variation in WBC counts, ranging from 0.5 to 16 × 10^6^/mL of semen. It is worth mentioning that many of these fertile men were within the 95th percentile range of the WHO criteria that define leukocytospermia (i.e., more than 1 × 10^6^ WBC/mL) [75]. On the other hand, within infertile men, the reported prevalence of leukocytospermia based on this cut-off value varies from 10% to around 20% [69,76]. These results show that leukocytospermia is not a strictly limiting factor for male fertility. In fact, Kaleli et al. [70] stated that “*leukocytospermia may have a favorable effect on some sperm functions at seminal leukocyte concentrations between 1 and 3 × 10^6^/mL*”.

Data regarding the impact of leukocyte on semen quality are always difficult to interpret because it is hard to predict: (1) when these cells had entered the seminal compartment; (2) whether they were activated; and (3) when and how they were activated. Normally, spermatozoa only encounter a large number of WBC upon ejaculation, and significant numbers of leukocytes are rarely seen in the lumina of the seminiferous or epididymal tubules [77]. In addition, upon ejaculation, when WBC generally contact sperm cells, seminal plasma is also present, thus protecting sperm with its antioxidant compounds [78,79]. Nevertheless, as soon as seminal plasma is removed, leukocytes may damage the spermatozoa, a tendency easily verified by the strong association between the presence of leukocytes in washed sperm preparations and in vitro fertilization (IVF) rates [80].

## 3. The Free Radical-Generating Systems in Sperm

One question still pending concerns the exact nature of the enzymatic systems responsible for free radical production in sperm cells (Figure 1). In a major review by Agarwal et al. [81], the authors indicate two ways spermatozoa may generate ROS, being: (1) an NADPH-oxidase system embedded in the plasma membrane [82]; and (2) an NADH-dependent oxidoreductase (diaphorase) at the level of mitochondria [83].

### 3.1. The Potential for an NADPH–Oxidase System in Sperm

The NOX hypothesis for sperm was conceived on the basis of two main observations. Firstly, ionophore A23187 was shown to increase the ROS-dependent chemiluminescent signal of either oligozoospermic samples [84] or capacitated sperm [85], indicating the action of a Ca^2+^-dependent NOX. Secondly, the addition of NAD(P)H to sperm suspensions can generate a dose-dependent increase in luminol–peroxidase signal and in nitro blue tetrazolium (NBT) reduction, indirectly suggesting O_2_•^−^ production [86,87]. In line with this theory, the NAD(P)H-dependent lucigenin signal was effectively inhibited by the addition of copper, zinc, diphenyleneiodonium (DPI) and SOD [86,87,88,89].

Following these previous observations, studies performed on equine and human sperm presented the NADPH-oxidase isoform 5 (NOX5) as one potential candidate for the ROS-generating system (Figure 1) [90,91,92]. This NOX isoform contains EF-hand Ca^2+^ binding domains, being activated by Ca^2+^ [90,93]. The mRNA expression of NOX5 is first detected in pachytene spermatocytes (human [90]), whereas the protein can be visualized in the developing spermatid (equine [91]). In human spermatozoa, a NOX5 antibody demonstrated cross-reactivity in the flagellum, neck and acrosome regions [92], with higher reactivity in asthenozoospermic men [94]. Additionally, Armstrong et al. [95] demonstrated that sperm NOX5 has a lower ROS-producing capacity when compared to WBC, and its activation is probably independent of protein kinase C. Despite these results, some points have not yet been satisfactorily addressed by previous reports, such as the possibility of leukocyte contamination in sperm samples, discrepancies in molecular weight and a lack of mass spectrometry evidence on the abundance of NOX5 in sperm [96,97,98]. Additionally, NOX5 is not found in rodents, which limits deeper pathophysiological studies.

Furthermore and in contrast to the results presented so far, the addition of NAD(P)H was also reported not to stimulate O_2_•^−^ production when the superoxide-dependent probe 2-methyl-6-(*p*-methoxyphenyl)-3,7-dihydroimidazo [1,2-a] pyrazine-3-one (MCLA) [99] and the electron spin method [100] were used, therefore questioning the existence of a NOX activity in sperm.

This contradiction was later elucidated when our laboratory successfully identified the enzymes responsible for the NAD(P)H-dependent lucigenin signal as cytochrome p450 reductase (CP450R) [101] and cytochrome b5-reductase (Cb5R) [102]. CP450R (acting preferably on NAD(P)H) and Cb5R (with higher affinity for NADH) are both capable of a direct one-electron reduction of either lucigenin (Figure 2) or tetrazolium salts (e.g., NBT and WST-1) (Figure 3), thus easily explaining why these probes can evoke a signal with NAD(P)H, whilst other methods had failed (i.e., MCLA and electron spin resonance). In addition, the reduction of lucigenin and tetrazolium salts by these enzymes forms unstable radicals that may also produce O_2_•^−^, which are essential for signal generation (Figure 2 and Figure 3; for more detail see [101] and [102]). Due to the latter, SOD has the ability to inhibit the NAD(P)H-dependent lucigenin chemiluminescence and the tetrazolium salt formation generated by CP450R and Cb5R (Figure 2 and Figure 3). Unfortunately, this inhibition by SOD is similar to the one expected when ROS is generated by NOX activity. Furthermore, like NOX, CP450R and Cb5R are also flavoproteins and, therefore, susceptible to DPI inhibition (Figure 2 and Figure 3). Taken together, these inhibition tests are not suitable to differentiate whether the lucigenin and the tetrazolium salt signals were generated by cytochrome and/or NOX activity.

### 3.2. Other Enzymatic Sources of ROS in Sperm

Given the compelling pieces of evidence supporting the importance of ROS in sperm physiopathology, studies are still needed in order to determine the involvement of other sperm enzymes in the production of O_2_•^−^. Besides NOX, the oxidative metabolism of AA, the second most abundant PUFA in human sperm cells [44], by cyclooxygenases (COX) and lipoxygenases (LOX) is also an important ROS-generating source. In this case, ROS can be generated as a by-product of AA oxidation [103] and/or as a result of NOX activation by either AA itself [104,105] or its LOX- and COX-generated metabolites [106,107] (Figure 1). The connection between LOX metabolites and ROS generation by NOX may be true for sperm cells. For instance, in mice germ cells, the inhibition of the isoform 15-LOX by PD146176 resulted in the reduction of ROS production within these cells [108]. Although not further investigated by Bromfield et al. [108], a study developed with Jurkat cells reported that 15-LOX metabolites may be involved in NOX stimulation [109].

### 3.3. Sperm Mitochondria and ROS Generation

In mammalian cells, another potential enzymatic source of ROS is the mitochondrial electron-transport chain. Under normal conditions, around 0.1–0.2% of the electrons passing the respiratory chain may leak and react with oxygen molecules, mainly forming O_2_•^−^ [110]. Electron leakage may occur in several sites within the respiratory chain, being the ubiquinone binding sites in complex I (Q-binding site; O_2_•^−^ is produced on the matrix side) and in complex III (Q_o_ site; O_2_•^−^ is produced in the intermembrane space) the most important ones [111,112] (Figure 1). In sperm, the specific inhibition of electron transport in complex I (by rotenone) and complex III (by antimycin-A) showed that these cells are also capable of producing ROS in these mitochondrial sites [113]. However, it is still unclear whether the ROS produced by mitochondria exerts specific physiological and/or pathological roles in sperm. For equine sperm, mitochondrial ROS were reported to positively correlate with sperm motility and velocity, probably due to an intense oxidative phosphorylation activity [114]. Nevertheless, contrasting with the equine species, human spermatozoa greatly rely on glycolysis for ATP production, with little contribution of oxidative phosphorylation [115]. For this reason, the interference of mitochondrial ROS in human sperm function may be less obvious. Of interest, defective human sperm have been shown to spontaneously generate mitochondrial ROS to a point that sperm motility may be affected [113].

## 4. ROS Measurement Techniques and Their Reliability

Throughout the literature on sperm and ROS, many statements and theories are still controversial and in need of re-examination and rectification. In part, this is due to some limitations and drawbacks which may be seen with the techniques and probes commonly used to evaluate ROS in living cells. Currently, no probe offers an unbiased measurement of ROS, with an ideal high reactivity and specificity for one ROS species. Importantly, this leads to a scenario in which unrealistic conclusions about the relationship between ROS and sperm function can be made. An extensive discussion on this theme can be found elsewhere (see [116]). In this review, we will limit the discussion to the probes that are more commonly used in the field of spermatology (Table 1).

### 4.1. Lucigenin and Tetrazolium Salts

The use of NAD(P)H in conjunction with either lucigenin or tetrazolium salt techniques has been previously discussed here in Section 3.1. In this case, the main concern is that several tissue reductases, including sperm cytochromes (CP450R and Cb5R) [101,102], can reduce both probes and, therefore, lead to artefactual NAD(P)H-dependent reduction and the generation of O_2_•^−^ by autoxidation [117,118] (Figure 2 and Figure 3). However, despite these consistent factors, many studies have used this approach to indirectly report the presence of O_2_•^−^ in sperm and further correlate it with semen quality [119,120,121], capacitation [122], hyperactivation [123], DNA integrity [120,124,125], apoptosis [120], IVF outcomes [121], among others. Yet, caution and a deep understanding of the limitations of both detection methods must guide the interpretation of these data.

### 4.2. Luminol/HRP

Luminol (5-amino-2,3-dihydro-1,4-phthalazine-dione) present the advantage of having a high sensitivity and the capacity to detect both intra- and extracellular ROS [82,118]. To react with O_2_•^−^, luminol is first converted into an intermediate radical by a one-electron oxidation normally mediated by H_2_O_2_ [126,127] and enhanced by the addition of horseradish peroxidase (HRP) [82,128] (Figure 4). One major limitation is the fact that the luminol radical reacts not only with O_2_•^−^ but also with various compounds capable of donating an electron [126,127], thus showing indiscriminate recognition of several free radicals. In addition, other complex and difficult-to-control factors, such as the formation of O_2_•^−^ by the luminol radical, may influence the chemiluminescence of this probe [118,127,129]. Therefore, according to Zhang and colleagues [118], it is “*unwise to monitor the dynamics of free radical generation in cells or systems with this probe alone*”.

Previous studies have used the luminol-based technique to suggest that pathological spermatozoa (e.g., amorphous heads, damaged acrosomes and retained cytoplasmic droplets) generate higher amounts of ROS than their normal counterparts [130,131]. Nevertheless, one possible interpretation for these data is that luminol–HRP reacts with sperm containing luminol-reactive metabolites not yet specified. This is reinforced by the fact that the retention of an excess of residual cytoplasm, a common feature of abnormal sperm, is associated with higher ROS measurements [132]. It is important to note that the excess of residual cytoplasm may contain higher amounts of the metabolites responsible for luminol signal, therefore not directly related to ROS production.

### 4.3. Dihydroethidium

Dihydroethidium (DHE), also called hydroethidine (HE), has been branded as a superoxide indicator and, when combined with the hexyl triphenylphosphonium cation (MitoSOX^TM^ Red), it can specifically detect mitochondrial ROS. Oxidation of DHE by intracellular O_2_^•^^−^ forms 2-hydroxyethidium (2-OH-E^+^), that emits a red fluorescence with excitation at 510 nm [133] (Figure 5). However, DHE is also susceptible to non-specific oxidation by other oxidants (e.g., H_2_O_2_, •OH), generating ethidium (E^+^), a compound with fluorescence characteristics similar to those of 2-OH-E^+^ [134]. For this reason, because both by-products of specific (2-OH-E^+^) and non-specific (E^+^) oxidation of DHE have overlapping fluorescence, quantification of O_2_•^−^ by this means is not possible when only fluorescence-based techniques are used. Of concern, many reports have used these methods to assess O_2_•^−^ in sperm cells, thus not considering the potential contribution of the non-specific oxidation of DHE via alternative pathways [135,136,137,138].

An alternative to unambiguously confirm the presence of intracellular O_2_•^−^ is to separately identify both 2-OH-E^+^ and E^+^ with techniques such as high-performance liquid chromatography (HPLC) and liquid chromatography–mass spectrometry (LC–MS) [139,140]. Using HPLC and a reversed-phase column, the 2-OH-E^+^ and E^+^ peaks can be separated and resolved, allowing O_2_•^−^ quantification [140]. To the best of our knowledge, this methodology has only been used to analyse menadione-treated spermatozoa [33] and has never been used to compare the level of ROS spontaneously generated by normal and pathological sperm cells. Recently, we have used the LC–MS/MS approach to investigate sperm O_2_•^−^ generation during in vitro incubation [141]. As previously reported, we also observed an increase in DHE over time. However, this was not accompanied by an increment in 2-OH-E^+^ levels but was rather a consequence of an increase in the level of E^+^ (i.e., not related to O_2_•^−^ generation). Our finding clearly shows the importance of distinguishing 2-OH-E^+^ from E^+^ when assessing O_2_•^−^ production.

## 5. Conclusions

From past to present, the knowledge gathered over the many years of study in this field offers us a few lessons that need to be taken into account. Firstly, to avoid any interference when assessing the production of ROS by sperm cells, an efficient removal of leukocytes from the samples is mandatory. Their presence will always cast doubt and potentially lead to data misinterpretation, as clearly evidenced by the work of Whittington and Ford [11]. Secondly, many of the methods used to assess ROS in sperm cells present drawbacks and limitations during application, possibly obfuscating the true nature of the involvement of free radicals in sperm physiology and male infertility. The rational use of probes and sometimes the adoption of more than one method are recommended for a better assessment of ROS in cells. An indirect assessment of oxidative stress may also be done by the analysis of the products originated from lipid (MDA, 4-HNE, HHE) [23,61,62,67] and DNA oxidation (DNA base adduct 8-hydroxy-2′-deoxyguanosine) [142,143,144].

Finally, although sperm are susceptible to in vitro induced and exogenous sources of ROS and its by-products, the in vivo relevance of these compounds needs further clarity. Of interest, considering only the ROS produced by sperm, our laboratory has recently found that neither O_2_•^−^ nor other free radicals, which lead to 4-HNE production, are responsible for motility loss during incubation [141]. In addition, a clear distinction must be made between the physiological versus the pathological roles of ROS in sperm. While a subtle increase in ROS may be necessary for sperm function such as in capacitation, the relationship between sperm abnormality and ROS may arise from a redox imbalance within the different environments to which sperm are subjected, especially in testis [145]. However, before any definitive conclusions are made, more studies using refined methodologies to look at the level of spontaneous ROS generation or lipid peroxidation in fertile and infertile males are required. In addition, while measurements of both 4-HNE and MDA have been performed in spermatozoa, the levels of 4-HHE, perhaps a more important aldehyde, still need to be evaluated.

## Figures and Tables

**Figure 1 antioxidants-08-00616-f001:**
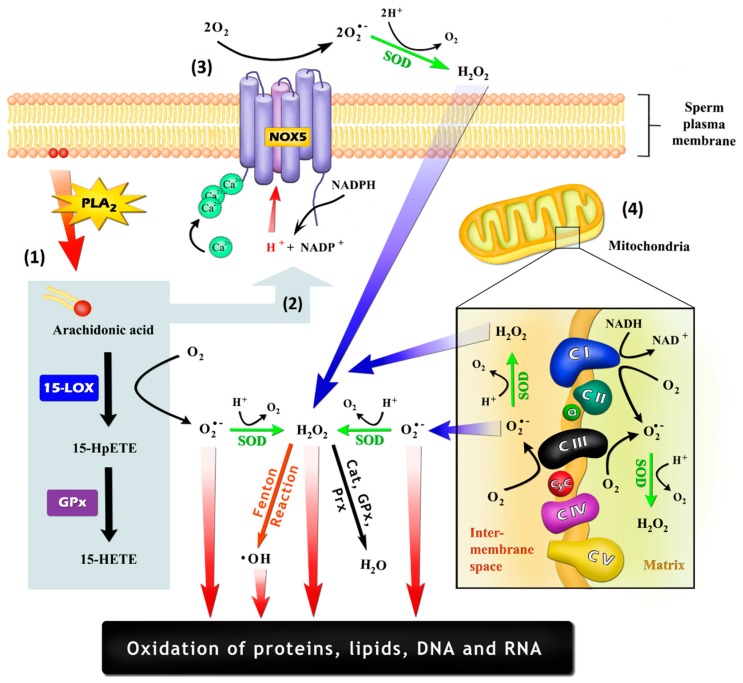
Possible mechanisms by which sperm cells may generate reactive oxygen species (ROS): (1) As a by-product of the oxidation of arachidonic acid (AA), which may be promoted by cyclooxygenases and lipoxygenases, such as arachidonate 15-lipoxygenase (LOX-15); (2) Through the stimulation of NADPH-oxidase (NOX) activity by AA itself or by their oxidation-generated metabolites, being 15-hydroperoxyeicosatetraenoic acid (15-HpETE) and 15-hydroxyeicosatetraenoic acid (15-HETE) potential inducers; (3) Generation by an NOX system, such as NADPH-oxidase isoform 5 (NOX5), which is embedded in the plasma membrane and is activated through an EF-hand Ca^2+^ binding domains; (4) Generation by the mitochondrial electron-transport chain, with the electron leakage within the ubiquinone binding sites in complex I (CI) and in complex III (CIII) being the most important mechanisms. Cat: catalase; CyC: cytochrome C; GPx: glutathione peroxidase; PLA2: phospholipase A2; Prx: peroxiredoxins; Q: ubiquinone; SOD: superoxide dismutase.

**Figure 2 antioxidants-08-00616-f002:**
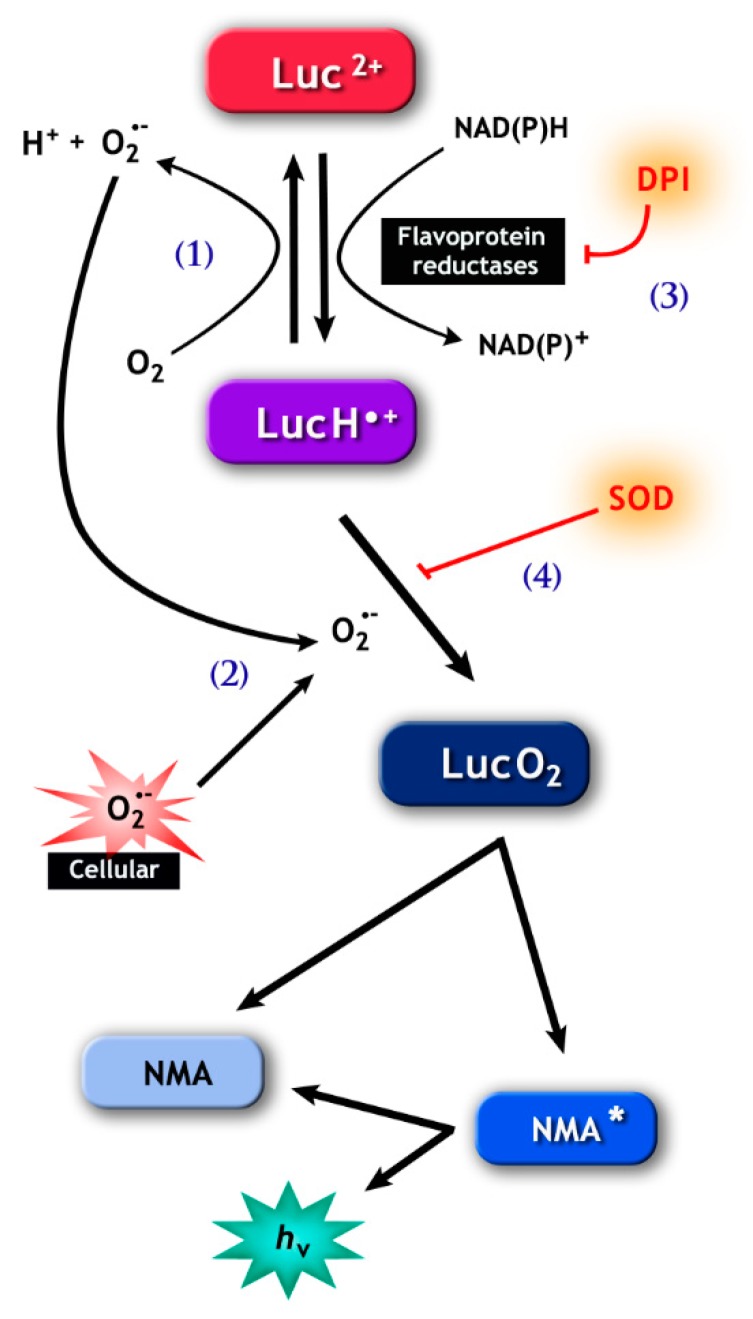
The pathways involved in lucigenin (bis-N-methylacridinium nitrate) chemiluminescence. (1), Lucigenin (Luc^2+^) can be reduced to the lucigenin cation radical (LucH•^−^) by flavoprotein reductases, including cytochrome b5-reductase (Cb5R) and cytochrome p450 reductase (CP450R). LucH•^−^ can be autoxidize back to lucigenin resulting in the production of (superoxide anion) O_2_•^−^, or (2) can react with O_2_•^−^, forming lucigenin dioxetane (LucO_2._) The latter spontaneously decomposes into N-methylacridone (NMA) that generates the chemiluminescence signal. Note that the signal can be abolished by diphenyleneiodonium (DPI) (3), which inhibits flavoprotein reductases, and by superoxide dismutase (SOD) (4), which consumes the O_2_•^−^ necessary for the formation of LucO_2_

**Figure 3 antioxidants-08-00616-f003:**
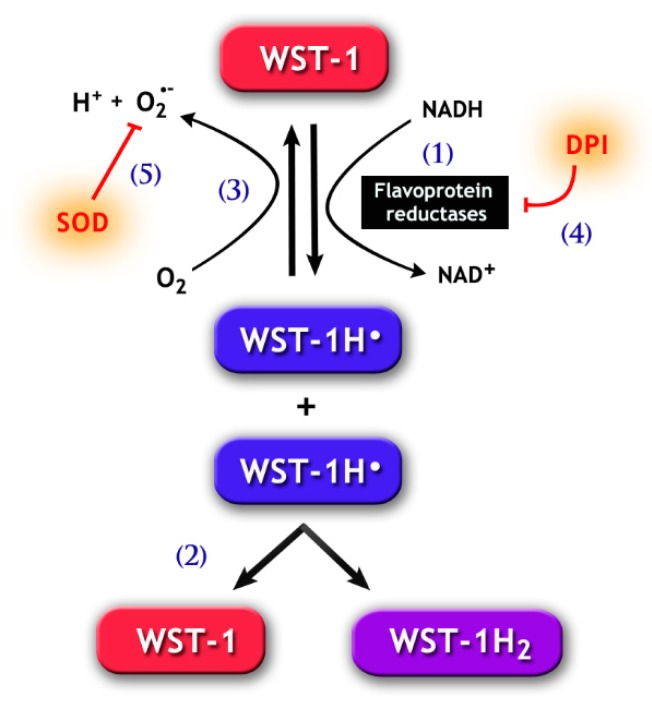
Chemical pathways for the 2-[4-iodophenyl]-3-[4-nitrophenyl]-5-[2,4-disulfophenyl]-2H tetrazolium monosodium salt (WST-1) assay. WST-1 can be reduced by electron transport from NADH via flavoprotein reductases, such as Cb5R and CP450R (1), forming the WST-1 radical (WST-1H•). The latter goes through disproportionation (2), which generates the reduced soluble purple formazan product (WST-1H_2_) detected by spectrophotometry methods. Notably, WST-1H• may also react with molecular oxygen, forming O_2_•^−^ (3). The generation of WST-1H• can be prevented by the addition of DPI (4), a flavoprotein inhibitor. Likewise, SOD (5) may inhibit the formation of WST-1H_2_, because the reduction of O_2_•^−^ concentration by SOD increases the autoxidation of WST-1H•, therefore reducing the latter’s availability for the formation of WST-1H_2_.

**Figure 4 antioxidants-08-00616-f004:**
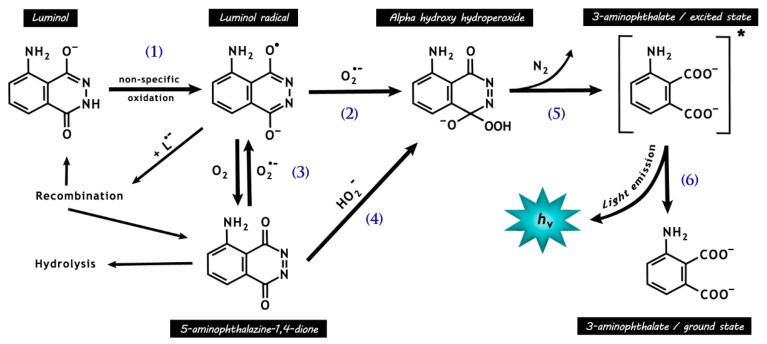
Chemical reactions responsible for luminol chemiluminescence. Luminol is first oxidized by many radicals (e.g., •OH and CO_3_•^−^, except O_2_•^−^) and peroxidases, forming the luminol radical (L•^−^) (1). L•^−^ then reacts with O_2_•^−^, forming the short-lived intermediate hydroperoxide (2). Molecular oxygen may be reduced to O_2_•^−^ by L•^−^ (3), with a rate around seven orders of magnitude lower than that for reaction (2), resulting in the production of 5-aminophthalazine-1,4-dione. The latter may also form the intermediate hydroperoxide by the addiction of hydrogen peroxide anions (4). The intermediate hydroperoxide is quickly decomposed to 3-aminophyhalane in an excited state (5), which emits light on relaxation to the ground state (6).

**Figure 5 antioxidants-08-00616-f005:**
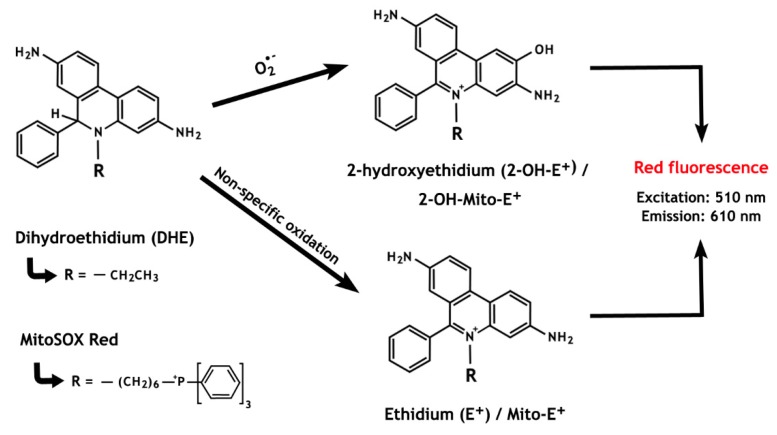
The chemistry behind the ROS detection methods based on dihydroethidium and MitoSOX Red oxidation. The non-specific oxidation, which forms ethidium, is predominant over the superoxide anion-induced reaction that results in the formation of 2-hydroxyethidium. Notably, both oxidized by-products present overlapping fluorescence properties.

**Table 1 antioxidants-08-00616-t001:** Characteristics and limiting factors of the probes commonly used to detect ROS in sperm cells.

Probe	Method	Characteristics and Limiting Factors
Tetrazolium salts	Colorimetric	Nitro blue tetrazolium (NBT) is the most commonly used one Low sensitivity to detect ROS Low specificity for O_2_•^−^ detection, with various intracellular reductases being able to generate the same response Autoxidation can generate O_2_•^−^
Lucigenin	Chemiluminescence	More specific for extracellular O_2_•^−^ Inability to detect O_2_•^−^ at low level Low specificity for O_2_•^−^ detection. Signal can be triggered by various nucleophiles and reducing agents, being sensitive to changes in the reductase activity within the tested systems. Reduced radical can generate O_2_•^−^
Luminol/HRP	Chemiluminescence	Allows the detection of both intra- and extracellular ROS Reacts with several electron-donor compounds, showing indiscriminate recognition of numerous free radicals The luminol radical formed by various univalent oxidants can form O_2_•^−^ through autoxidation Susceptible to various interferences in biological systems, such as poor ROS detection at neutral pH and absorption of the emitted light (400 nm) by some biomolecules
DHE	Fluorescence HPLC and LC–MS	Used to detect intracellular O_2_•^−^ Highly specific for O_2_•^−^ detection, producing 2-hydroxyethidium (2-OH-E+); however, the majority of DHE reacts with other oxidants, resulting in the production of ethidium (E+) Both by-products of non-specific (E+) and specific (2-OH-E+) oxidation have overlapping fluorescence properties, thus not allowing distinction by fluorescence methods. For specific O_2_•^−^ quantification, 2-OH-E+ must be measured by techniques such as HPLC and LC-MS

Dihydroethidium (DHE); high-performance liquid chromatography (HPLC); horseradish peroxidase (HRP); liquid chromatography–mass spectrometry (LC-MS); superoxide anion (O_2_•^−^).

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
