# Peer review of "From Past to Present: The Link Between Reactive Oxygen Species in Sperm and Male Infertility"

_antioxidants, 2019, doi:10.3390/antiox8120616_

Round 1

Reviewer 1 Report

The manuscript entitled “From Past to Present: the Link between Reactive Oxygen Species in Sperm and Male Infertility” reviews the mechanisms of ROSs production, their measurement techniques, and their roles in sperm function. Despite of comprehensive reviews, the manuscript requires major revision on organization and scientific writing. Also, the manuscript does not fit the title with its objective. To be published the manuscript needs major revision.

More detailed technical comments have been provided below.

Major comments

The objective of this review is to discover the novel and more efficient method to measure ROS together with a better understanding of the pathways associated with peroxidative damage. Therefore, authors should change the title to suit for their objective of this review. In addition, this manuscript mostly contains the relation between ROS and sperm motility, and it would be better to include the another sperm functional parameters and the relationship of ROS. It would be a better compare and review if the authors consider the addition of sensitivities or mechanisms of ROS in aspect of animals, not just in human cases for greater impact. Therefore, to do this, substantial references of animal studies should be cited. All individual sections, including ‘INTRODUCTION’, are too brief. The author seems to be writing in chronological order, but Introduction still doesn’t contain necessary information. Also, each subsection does not contain enough content to explain it. If the author considers to re-organize the pathophysiology of ROS, the manuscript would be greater. All the figures are very good at understanding the mechanism of ROS, but figure legend is too simple. Please rephrase the figure legends for better explanation. Even though one of the objective of this manuscript is to find out the novel and improved methods for measuring ROS, it is too difficult to get this kind of information from this review. Therefore, to make it easier for readers to get information, it is better to provide a table comparing the advantages and disadvantages of each method.

Minor comments

Though the writing of this manuscript is able to be accepted, it still needs to be revised further. It is suggested to ask for an English native speaker to conduct a comprehensive editorial reading and revising.

Author Response

We thank the reviewers for their comments. Review comments are in boldface, response is in Italic.

Reviewer 1

Major comments

The objective of this review is to discover the novel and more efficient method to measure ROS together with a better understanding of the pathways associated with peroxidative damage.

We feel the reviewer is close to our objective, but perhaps we can also reiterate. The review is an objective and honest appraisal of where the field of ROS and sperm research is at the present moment. Currently, the foundation of this research has many “cracks” and this is due to the fallacious assays that have/are being used. Our objective is to strengthen the foundations better and make sure the field understands there are alternative explanations for many of the data that have been ONLY interpreted to be an effect of ROS. We must learn historically where things need strengthening in order to get to the truth.

Therefore, authors should change the title to suit for their objective of this review.

We can change the title if the reviewer insists, however, it does reflect the review as it stands. Our review goes into details in regards to the past and present data and interprets this in the context of male-factor infertility. If the reviewer has an alternative title, we would be happy to consider this.

In addition, this manuscript mostly contains the relation between ROS and sperm motility, and it would be better to include the another sperm functional parameters and the relationship of ROS.

The majority of research done on the field of ROS and male-infertility has dealt with motility issues. We point out that most of this work is based on addition of exogenous agents that cause ROS, which in turn affect motility. There is some data on “ROS” generation and morphology, however, we make a clear point that all of these assays should not be used (for example, NAD(P)H-lucigenin, Luminol-HRP). As such, we do not want to say (nor should it be said), that the use of these assays link “ROS” and poor sperm morphology. We actually make a point that the use of the above assays may link sperm morphology to excess residual cytoplasm, since the contents of the cytoplasm would ensure a greater signal in both assays.

It would be a better compare and review if the authors consider the addition of sensitivities or mechanisms of ROS in aspect of animals, not just in human cases for greater impact. Therefore, to do this, substantial references of animal studies should be cited.

This point really comes back to the main objective which is why we have reiterated our objective. That is, to document historically assays and interpretations of “ROS” to male-infertility. We are not focused on animal studies (although several have been cited).  Please note, we have ~150 references cited in this review, more than what a typically review would cite.

All individual sections, including ‘INTRODUCTION’, are too brief. The author seems to be writing in chronological order, but Introduction still doesn’t contain necessary information. Also, each subsection does not contain enough content to explain it.

The theme of ROS in biology and even in spermatology is very vast. For the present review, we want to propose a different view of the field; how some of the principles were constructed during the many years of research and reasons to adopt a careful scrutiny and interpretation of some the available data. Regarding the feeling of chronology, it was in fact something intended by the authors (as stated in the title), and we are pleased to see that it was somehow achieved.

            It is worth noting that this review is written to a specialised audience, and many of the arbitrary details may have been missed out since it was not our intention to make the review unnecessarily long. Nevertheless, it’s a very detailed review with every major article that changed the field of ROS cited. We had other people reading through the manuscript and everyone seems to be able to understand it. If the reviewer insists on re-writing some section, perhaps be a little more informative as to what information need to be further complemented.

If the author considers to re-organize the pathophysiology of ROS, the manuscript would be greater.

It’s very unclear to us what “pathophysiology of ROS” would mean in the context of our review. For example, the only two definitive ROS measurements (electron spin and DHE with HPLC separation) are highly inconclusive. For this reason, before we start drawing conclusions on “pathophysiology”, we have to agree on whether “ROS” has been measured at all. In most of the cases, we point out that ROS is unlikely to have been measured. Secondly, the only “pathophysiology” we find is when researchers exogenously add ROS to sperm, and then motility is lost. This point was well discussed. This review shows that we need better assays and understanding of measurements before conclusions on ROS can be put forward.

All the figures are very good at understanding the mechanism of ROS, but figure legend is too simple. Please rephrase the figure legends for better explanation.

We have re-phrased the figure legends.

Even though one of the objective of this manuscript is to find out the novel and improved methods for measuring ROS, it is too difficult to get this kind of information from this review. Therefore, to make it easier for readers to get information, it is better to provide a table comparing the advantages and disadvantages of each method.

We have added a table. We understand the reviewers point but the aim of the review is to show how NOT SUITABLE many of the methods are.  The only suitable ones we are aware of are DHE (with HPLC separation) and electron spin (bearing in mind that the latter shows ROS produced on sperm, due to insensitivity).

Minor comments

Though the writing of this manuscript is able to be accepted, it still needs to be revised further. It is suggested to ask for an English native speaker to conduct a comprehensive editorial reading and revising.

The manuscript has been through 4 native English tongues, including an Englishman and several Australians. In addition, the non-native English speaker included in the manuscript has proven experience in paper writing. In addition, reviewer 2 has praised the manner in which the review was written. Therefore, perhaps this comment is more related to a writing style discomfort. Perhaps, the reviewer could be more specific. In any case, some minor changes were made.

Reviewer 2 Report

In this Review article the authors critically discuss some literature regarding reactive oxygen species (ROS) and its possible production/effects on mammalian sperm, function and male infertility. Several aspects are covered, including historical observations and their possible significance, as well as potential mechanisms of ROS production, reasons why sperm may be affected, and some of the different probes/methods used to monitor ROS, as well as their possible technical drawbacks.

Overall, and although clearly biased towards the authors own interests (which is normal, and to be expected)  this is a very well organized and written Manuscript, and interested readers will especially enjoy the historical framework and the many important issues the authors raise with both the biochemical aspects of ROS formation and potential action in this specific context, as well as with issues related to measurements.

There are however some aspects that the authors should consider in a revised version.

1- From reading the Manuscript (Abstract, Introduction, and elsewhere) it seems that almost all studies, notably in human sperm, involve artificial in vitro produced ROS, thus resulting in data that may have little physiological relevance. While there is certainly a point to this, there are also many many studies in which ROS are measured in human sperm from a variety of patient types and samples (not necessarily for studies in ROS, but as an aditional parameter for sperm function), and where any artificial ROS production is merely used as a positive control, to ensure that the probes are working. These studies should also be reviewed in this Manuscript in some way, in my opinion. I believe that in this particular stage the literature is not well represented by the authors at all, and this should be revisited. Of course there are other issues to possibly discuss in this case, besides the probes themselves (the role of sperm preparation, centrifugations, media changes, in vitro incubations needed for probe internalization, etc.), but the way the whole review is framed on the onset is not appropriate, in my view.

2- In the conclusions the authors mention some of their own unpublished data, which cannot, obviously, be critically evaluated at this stage. I believe it would be more correct to remove this reference, or rephrase the sentence.

Author Response

We thank the reviewers for their comments. Review comments are in boldface, response is in Italic.

Reviewer 2

In this Review article the authors critically discuss some literature regarding reactive oxygen species (ROS) and its possible production/effects on mammalian sperm, function and male infertility. Several aspects are covered, including historical observations and their possible significance, as well as potential mechanisms of ROS production, reasons why sperm may be affected, and some of the different probes/methods used to monitor ROS, as well as their possible technical drawbacks.

Overall, and although clearly biased towards the authors own interests (which is normal, and to be expected)  this is a very well organized and written Manuscript, and interested readers will especially enjoy the historical framework and the many important issues the authors raise with both the biochemical aspects of ROS formation and potential action in this specific context, as well as with issues related to measurements.

There are however some aspects that the authors should consider in a revised version.

From reading the Manuscript (Abstract, Introduction, and elsewhere) it seems that almost all studies, notably in human sperm, involve artificial in vitro produced ROS, thus resulting in data that may have little physiological relevance. While there is certainly a point to this, there are also many studies in which ROS are measured in human sperm from a variety of patient types and samples (not necessarily for studies in ROS, but as an aditional parameter for sperm function), and where any artificial ROS production is merely used as a positive control, to ensure that the probes are working. These studies should also be reviewed in this Manuscript in some way, in my opinion. I believe that in this particular stage the literature is not well represented by the authors at all, and this should be revisited.

We appreciate this appraisal of the review. This is in fact a very good point, and in some parts of the review we invite readers to think about the real physiological role of ROS in sperm cell. One major issue here is that in many of these studies inappropriate measurements of ROS were used. For example, early studies used thiobarbituric acid to measure MDA and found men with poor morphology have higher levels of “MDA”. However, we point out that such an assay measures many different types of compounds and you cannot assume that it is all ROS-mediated (ROS produced aldehydes which then react). We have tried to show the biochemistry behind the probes. As such, when readers go through different papers, they can then “judge” for themselves if an appropriate assay was used or not. In this regard, we even added a new table for the manuscript, as kindly suggested by reviewer 1. Nevertheless, even with more appropriate techniques, careful interpretation of this kind of study is recommended. Note that findings can be circumstantial and perhaps related to a redox-imbalanced milieu during spermatogenesis and maturation and not necessarily related to a ROS production by spermatozoa. Furthermore, even in the current literature it is not uncommon to find controversies regarding sperm function/morphology and ROS measurement.

Of course there are other issues to possibly discuss in this case, besides the probes themselves (the role of sperm preparation, centrifugations, media changes, in vitro incubations needed for probe internalization, etc.), but the way the whole review is framed on the onset is not appropriate, in my view.

Indeed, many factors can interfere with the assays. For example, medium pH (information in Table 1) and the presence of antioxidants directly affect luminol signal. In the review, we even mention about the importance of withdrawing white blood cells from sperm samples and how it can interfere with results. We also mention that “Early studies with rabbit, mouse and human sperm have shown that spontaneous lipid peroxidation, based on MDA measurements, occurs at a slow rate and factors such as temperature, oxygen tension and medium composition may greatly interfere [37-40].” However, although attention to sample preparation are mandatory, there is still a scarcity of trials in this regard. In addition, we are still in the phase of trying to show that assays and probe options need to be reviewed. If one probe is not suitable due to their biochemistry nature, it seems not reasonable at this stage to discuss how sample preparation may interfere with its efficiency.

In the conclusions the authors mention some of their own unpublished data, which cannot, obviously, be critically evaluated at this stage. I believe it would be more correct to remove this reference, or rephrase the sentence.

The paper is now accepted  and we have added the citation. As such, we will leave the comments as they stand.

Round 2

Reviewer 1 Report

The manuscript has been revised substantially according to reviewer's comments. 

Reviewer 2 Report

The authors remain, in my opinion, very biased, and have basically made minor changes, if any  at all, which suggests they won't regardless.

But the paper is of interest, as I said previously, and can be judged critically by any committed readers.